# Transformational Collaboration for the SDGs: The Alianza Shire's Work to Provide Energy Access in Refugee Camps and Host Communities

**Jaime Moreno-Serna** [1,*], **Teresa Sánchez-Chaparro** [1,2], **Javier Mazorra** [1], **Ander Arzamendi** [1], **Leda Stott** [1] **and Carlos Mataix** [1,2]

[1] Centro de Innovación en Tecnología para el Desarrollo Humano, Universidad Politécnica de Madrid (itdUPM), 28040 Madrid, Spain; teresa.sanchez@upm.es (T.S.C.); javier.mazorra@upm.es (J.M.); ander.arzamendi@upm.es (A.A.); leda.stott@upm.es (L.S.); carlos.mataix@upm.es (C.M.)

[2] Department of Organizational Engineering, Business Administration and Statistics, Escuela Técnica Superior de Ingenieros Industriales, Universidad Politécnica de Madrid, 28006 Madrid, Spain

\* Correspondence: jaime.moreno@upm.es; Tel.: +34-910-6713-97

**Abstract:** The potential for achieving transformation through partnerships is central to the 2030 Agenda for Sustainable Development. However, information on experiences that explore the processes that might generate systemic change is generally lacking. This article uses the collaborative value creation (CVC) framework to analyze the transformational prospects of the Alianza Shire, the first multi-stakeholder partnership for humanitarian action in Spain. The partnership, which aims to develop innovative energy access solutions in refugee camps situated in the Shire region of northern Ethiopia, is assessed from its creation in 2014 to the present, with regard to four key partnership features: organizational engagement, resources and activities, partnership dynamics, and impact. Our findings suggest that while the CVC framework is a useful tool for analyzing the evolution of a partnership to a transformative phase, additional information is required on the important role played by a partnership facilitator in assisting this process. This inquiry aims to build upon the CVC analysis by identifying and addressing some of the barriers faced by the Alianza Shire and other partnerships in attaining transformational outcomes and proposing two key enablers that can assist progression towards this: a facilitating organization that ensures the creation of collaborative shared value and an aspirational strategy for achieving significant systemic change.

**Keywords:** multi-stakeholder partnerships; collaborative value; refugees; energy access; intermediary; transformation.

## 1. Introduction

In 2015, the United Nations General Assembly adopted the 2030 Agenda for Sustainable Development and its 17 sustainable development goals (SDGs). One of the SDGs, Goal 17, focuses on strengthening and revitalizing a global partnership for sustainable development. Target 17.17 further endorses the encouragement and promotion of "effective public, public–private, and civil society partnerships" that build upon partnership experience and resourcing strategies [1]. This call has been supported by wider claims that multi-stakeholder partnerships offer the institutional and organizational structures needed to foster systemic transformation at the heart of the SDGs [2].

Several scholars defend the need for transformative approaches to address the challenges posed by the SDG Agenda [3,4]. According to Horan (2019), the premise is that "these transformations seek to exploit synergies between Goals to achieve multiple SDGs by organizing implementation around SDG interventions that generate significant co-benefits". In this regard, multi-stakeholder

partnerships are viewed as vehicles that can help to accelerate these synergies and build more enduring governance structures [4–6].

The refugee crisis has been described as a challenge that requires a transformation in the international community's response to forced displacement [7]. According to data from the United Nations High Commissioner for Refugees (UNHCR), by the end of 2015, there were more than 55 million displaced people around the world, 14 million of whom were refugees forced to flee due to persecution, conflict, repression, and natural disasters. At the end of 2018, the number of forcibly displaced individuals worldwide had increased to almost 70.8 million, 26 million of whom were refugees [8]. The humanitarian community's priority is to provide these people with basic services such as shelter, food, water, and protection. In spite of its relevance and cross-cutting impact, access to energy has generally been disregarded when considering the needs of refugees. However, in recent years, several international initiatives have been established to address this problem, including the Moving Energy Initiative (MEI) [9], Safe Access to Fuel and Energy (SAFE) [10], and the Global Plan of Action for Sustainable Energy Solutions in Situations of Displacement [11].

The humanitarian community has called for the active implication of the private sector in the response to the refugee crisis [7,12]. Traditionally seen as little more than an alternative source of funding, the private sector has increasingly been playing other roles in the humanitarian sector, most notably in product and process innovation [13–15]. Partnerships between private companies, national and international agencies, academy, NGOs, and refugees appear to offer more inclusive planning and designing approaches [16] and market-based strategies can underpin initiatives such as access to energy in fragile places, making them more effective and durable [17–19].

Against this background, and in recognition of the potential for engagement of several Spanish electricity and energy companies, the Spanish Humanitarian Action Office and the Innovation and Technology for Development Centre at the Universidad Politécnica de Madrid (itdUPM), began to explore the possibility of launching a multi-stakeholder partnership to develop innovative energy access solutions to improve the services and quality of life of those living in refugee camps situated in the Shire region of northern Ethiopia. As a result, in 2014, five entities from the private, public, and academic spheres created the first multi-stakeholder partnership for humanitarian action in Spain: the Alianza Shire [20]. The partnership currently works to provide energy access to 60,000 refugees in the camps and 17,000 people from the host communities in the Shire region through a project that focuses holistically on the main sustainability drivers for this kind of intervention: technology quality assurance, assessment of community payment capacity, operation and maintenance, supply through business models, and influence on public policies [21].

Within the partnership, itdUPM, which was set up as an organizational laboratory to accelerate solutions that support transformative partnerships for the SDGs [22], adopted a proactive facilitation role by seeking to overcome barriers that might hinder collaboration between the partner organizations. This is a role that many authors recognize as necessary, especially in the humanitarian field where the private sector, particularly, has sometimes had difficulties in understanding contextual complexities and the many actors involved [13,23–25].

The motivation for this article is the internationally recognized need for transformative partnerships that can provide more effective and durable responses to forced displacement. An illustrative case study is offered on how collaboration among different organizations from the profit, non-profit, public, and academic sectors, seeks to provide energy access to refugee camps.

The collaborative responses required for multi-stakeholder partnerships such as the Alianza Shire are likely to rest upon the generation of meaningful shared value that promotes transformation and systems change. Shared value, a concept coined by Porter and Kramer (2011) [26], can be defined as policies and operating practices that enhance the competitiveness of a company while simultaneously advancing the economic and social conditions in the communities it operates. Porter and Kramer (2011) further suggest that multi-stakeholder collaboration, among other strategies, can enable companies to create shared value through the development of industry clusters among both private companies and other actors.

Austin and Seitanidi (2012) [27] focused on the potential of collaboration for generating shared value. In particular, they proposed a collaborative value creation (CVC) framework for analyzing value creation in relation to different types of collaborative relationships between private and non-profit organizations [27]. Austin and Seitanidi conceptualized an advanced stage of collaboration which they call "transformational collaboration". This stage represents collaborative social entrepreneurship that "aims for value in the form of large-scale, transformational benefit that accrues either to a significant segment of society or to society at large". According to Austin and Seitanidi (2012), there is a need to deepen our understanding of the factors that enable collaborative relationships to enter into a transformational stage. Given that these forms of partnership are both complex and poorly documented, they called for the development of in-depth case studies to further explore their transformational potential.

Although not considered explicitly by Austin and Seitanidi (2012), partnership facilitators (also known as partnership brokers or partnership intermediaries) are increasingly acknowledged for their ability to promote transformation by shaping, nurturing, and supporting innovative collaborative relationships as they evolve over time [4,28–34]. This function may be assumed by an individual or an organization, and sometimes both [28,35]. The role of intermediary actors in facilitating transformation towards sustainability has been particularly noted in the sustainability transitions theory [36–38], with recognition of their function as key catalysts that speed up change towards more sustainable socio-technical systems [39]. Previous studies have suggested that this intermediary role could be played by different actors, including NGOs, public agencies, local authorities, or universities [40–42]. Because of their potential for offering spaces that encourage inter-disciplinary and multi-actor collaboration, universities and research centers such as itdUPM appear particularly well-suited to this intermediary role [41]. Further research is needed in order to fully understand the role of intermediaries in facilitating sustainability transformations and, in particular, in fostering multi-stakeholder partnerships [28,39] as well as the type of organizations that could fulfill this role.

This paper uses the Alianza Shire as an illustrative case study of a multi-stakeholder partnership with a clear transformational aspiration in order to begin to address the two research gaps identified above—improving our understanding of transformational multi-stakeholder collaborations and contributing to more deeply understanding the role of intermediaries in supporting these relationships. The CVC framework provides a useful lens for examining the extent to which the Alianza Shire has been able to "co-create significant economic, social, and environmental value for society, organizations, and individuals" [26] in relation to two key features: the transformational character of the partnership and its evolutionary nature [4,6]. The Alianza Shire is analyzed in its evolution from a preliminary prototype phase to one that has grown in scale and impact. Particular attention is paid to the role of a university center, itdUPM, in relation to the CVC spectrum and the extent to which its practical experience of promoting and facilitating collaboration within the Alianza Shire has supported progress towards transformation.

The article is organized as follows: Section 2 provides details of the theoretical background to the article. Section 3 presents the research methodology used which includes a promising framework for offering insights into how to promote collaborative value creation from an evolutionary perspective (CVC) using the consolidated methodology of a case study. In Section 4, an overview of the Alianza Shire is provided in order to assist a better understanding of the subsequent analysis. In Section 5, the evolution of different key parameters in relation to the Alianza Shire's potential for collaborative value creation is presented with analysis of the role that itdUPM has played in this evolution as a facilitating organization. Key findings are provided in Section 6 with a discussion of how the CVC framework perspective has helped itdUPM to identify barriers that hinder collaboration and how these lessons may be useful for other partnerships seeking to accelerate transformations in order to achieve the targets of the SDGs.

## 2. Theoretical Context of Partnerships

Collaborative arrangements that combine the resources, skills, and competencies of different actors in society in new and innovative ways are increasingly being proposed as vehicles for achieving transformation through partnerships for the SDG agenda [2,3]. However, information on the processes that might foster the full and meaningful participation of different stakeholders in these initiatives in order to generate systemic change is noticeably lacking [4–6,39].

Partnership stakeholders will assume different levels of power and risk in relation to the contributions they make to a collaborative initiative and to their interest in it. It is useful here to distinguish between internal and external stakeholders in a partnership. Stott (2009) classifies internal stakeholders as recognized signed-up partners who contribute resources to a partnership, assume risk on its behalf, have an important stake in how the relationship evolves, and stand to gain benefits from their involvement in it [43]. As a result of their position, we may thus expect each partner to assume a central role in collaborative decision-making. In spite of an emphasis on shared ownership, however, power dynamics between different partners are likely to exist, particularly in relation to perceptions of the value of different contributions, with financial input or political influence, for example, often being more highly esteemed than other resources. External stakeholders, meanwhile, are not signed-up partners of a collaborative initiative. They may include "interested observers" such as donors, media, or public authorities, who exert influence upon a partnership through financial resources, communication reach, and political leverage (or the promise of them); and "risk bearers" such as local actors or community players who may have weaker influence on a partnership but nonetheless bear important risks in relation to it [43].

Because partnership processes are conditioned by specific and changing contexts, the stakeholders described above may change their positions during the lifetime of a partnership [43]. As well as reflecting wider societal power relations in particular contexts, partnership relationships will also have their own "chemistry" and "distinctive 'arenas of power' where the emphasis on participation and consensus shapes power relations in particular ways" [44]. In addition, as stakeholders may have different interests in partnership at different times in different contexts, investment in efforts that seek to address inequitable power relations and ensure "appropriate" participation for diverse actors during a partnership's development are recommended [45].

"Appropriate" participation rests crucially upon ensuring clear and inclusive partnership governance mechanisms with horizontal decision-making processes that seek to minimize inequitable power relationships and promote shared ownership [46,47]. Caplan (2005) [48] believes that effective collaborative governance requires that partners demonstrate they are accountable to both each other and to external stakeholders. He suggests that partnership accountability requires systems and procedures that ensure (1) compliance, so that partners can hold one another to account; (2) transparency, where partners give each other an account of activities and progress; and (3) responsiveness, by which partners show that they have taken account of each others' needs or concerns, and those of wider stakeholders [48]. Caplan (2003) further notes that rigidity in pursuing accountability can limit space for innovation and that, "The key challenge for partnership practitioners is to strike an appropriate balance between formal structures that guide working practices, and leadership (in its broadest sense) that promotes creative thinking on how to maximize the inputs from different organizations" [49].

In most partnership arrangements, an important role in encouraging equitable participation and accountable governance processes is played by a partnership facilitator. This "partnership broker" [35] function may be assumed by an individual or an organization, and sometimes both, operating from both within or outside a partnership arrangement to shape and manage collaborative processes [28,35]. According to Manning and Roessler (2013) [50], the work of these "bridging agents" "who interact across multiple boundaries and translate ambiguous conditions into collaborative opportunities and constraints" is central to the development of effective long-term partnerships. By assisting partners and wider stakeholders to understand and respond to challenges, and providing spaces for to debate and reflection upon the way that they work together, partnership intermediaries

are instrumental to ensuring that collaborative efforts are inclusive in their approach, able to adapt to changing circumstances, and achieve sustainable results [51].

Horan (2019) notes that "several studies call for an *orchestrator* of partnerships" [4]. […] Most studies view orchestration as initiating and supporting individual partnerships. Proposed orchestrators include international institutions, government departments [43–45], or professional orchestrators [46]". As multi-actor partnerships are now widely promoted as vehicles for achieving the transformational agenda of the SDGs [1,2], understanding how partnership intermediaries support approaches that contribute to systemic change and the achievement of "significant economic, social, and environmental value for society, organizations, and individuals" [27] is of growing interest [4,6,28].

The role of intermediary actors in facilitating transformation for sustainability has also been explored in the sustainability transitions (ST) theory. This theory aims to conceptualize and explain how systemic socio-technical changes occur to address complex sustainability problems such as those represented in the SDG agenda [36,37]. ST theory, particularly the so-called "multilevel perspective", suggests that sustainability transitions take place at the interplay between three analytical levels: niches (the locus for radical innovations), socio-technical regimes (the locus of established practices and associated rules that stabilize existing systems), and an exogenous socio-technical landscape [36]. Within this body of knowledge, intermediary actors have been positioned as key catalysts that speed up change towards more sustainable socio-technical systems [39].

To explore the potential for transformation that may be derived from working in collaborative arrangements, Austin and Seitanidi (2012) propose a collaborative value creation (CVC) framework [27]. The premise of the CVC framework is that the main justification for cross-sector partnering is value creation among both for-profit and non-profit organizations. The concept of *Shared Value* is central to the CVC framework. First proposed by Porter and Kramer (2011), the notion of shared value rests upon the assumption that working with other actors, for-profit organizations can "develop policy and practices that enhance the competitiveness of a company while advancing the social conditions of the communities in which it operates" while non-profit organizations simultaneously have the "opportunity to be more effective, thinking in value terms: considering benefits relative to costs rather than funds and efforts expended" [26].

Although widely used and recognized, the notion of shared value has been criticized for, among other issues, its narrow private sector focus, failure to adequately address tensions between economic and social objectives, and lack of linkages to social innovation [52,53]. The CVC framework addresses some of these criticisms by providing a set of tools which conceptualize key elements and processes in fostering shared value for both profit and non-profit organizations through cross-sector partnerships. The CVC spectrum furnishes new reference terms for defining and analyzing value creation and proposes a continuum through which partners can enhance the generation of meaningful shared value by reviewing their work in relation to four stages of collaboration: philanthropic (a charitable relationship based on unilateral transfer of funds from one party to another), transactional (involving "reciprocal exchange" of resources through activities such as sponsorships and personal engagements), integrative (where value is created by combining key distinctive competencies and resources), and transformational (where value is derived from innovative co-creation processes between partners so that "transformative effects would not only be in social, economic, or political systems but also change each organization and its people in profound, structural, and irreversible ways") [27]. The last two components of the framework relate to partnering processes that reveal value creation dynamics in formation and implementation stages, and collaboration outcomes which examine impact at the micro, meso, and macro levels.

Rather than presenting a linear or fixed lens for analysis, Austin and Seitanidi [27] reinforced the importance of using the CVC continuum as a way of exploring evolving collaborative relationships. They proposed the use of a series of key pointers for this purpose (see Figure 1): level of engagement, importance to mission, magnitude of resources, type of resources, scope of the activities, interaction level, trust, internal change, managerial complexity, strategic value, co-creation of value, synergistic value, innovation, and external system change.

| NATURE OF RELATIONSHIP | Stage I: **PHILANTROPIC** | Stage II: **TRANSACTIONAL** | Stage III: **INTEGRATIVE** | Stage IV: **TRANSFORMATIONAL** |
|---|---|---|---|---|
| Level of engagement | Low | | | High |
| Importance to mission | Peripheral | | | Central |
| Magnitude of resources | Small | | | Big |
| Type of resources | Money | | | Core competences |
| Scope of activities | Narrow | | | Broad |
| Interaction level | Infrequent | | | Intensive |
| Trust | Modest | | | Deep |
| Internal change | Minimal | | | Great |
| Managerial complexity | Simple | | | Complex |
| Strategic value | Minor | | | Major |
| Co-creation of value | Sole | | | Conjoined |
| Synergistic value | Occasional | | | Predominant |
| Innovation | Seldom | | | Frequent |
| External system change | Rare | | | Common |

**Figure 1.** Variables used to characterize the evolutionary nature of partnerships in the CVC framework (source: Austin and Seitanidi, 2012).

Although the CVC framework expands upon Porter and Kramer's notion of shared value, its categorization of actors into "nonprofits and businesses" continues to transmit a somewhat limited vision of the diverse nature of the actors who may be involved in different collaborative initiatives. As a result, the creativity and innovation that may emerge from working with multiple stakeholders from the public, private, and civil society sectors at different levels, and the manner in which these unique connections, including the challenges and tensions that they generate, may encourage a move towards transformation is overlooked. The "orchestration" of such dynamics and the role exercised by a partnership intermediary in shaping and facilitating transformational relationships is also absent. Finally, while the CVC framework offers a useful conceptual tool for analyzing how shared value can be created through collaboration in relation to different evolutionary stages, it is, as the authors acknowledge, clear that it requires further testing in relation to concrete examples of partnerships in practice. In view of the SDG Agenda's emphasis on the importance of transformation, the need for detailed case studies that explore and share information on the process of building multi-stakeholder partnerships that seek to promote systemic change is particularly pressing.

## 3. Research Approach

### 3.1. Research Aims and Scope

This paper presents the evolution of the Alianza Shire since its creation and analyses its transformational potential through the lens of the CVC framework. The objective of the study is twofold:

- To analyze the evolution of the transformational character of the Alianza Shire and evaluate its position within the CVC spectrum.
- To characterize the role of Universidad Politécnica de Madrid's Innovation and Technology for Development Centre (itdUPM) as an intermediary actor in fostering the transformational character of the Alianza Shire through the identification of key intermediary activities.

Our work has both theoretical and practical implications. From a theoretical point of view, this case study aims to improve our understanding of transformational multi-stakeholder collaborations and contribute to clarifying the role of intermediaries. From a practical point of view, the analysis of

the Alianza Shire through the CVC lens enables the extraction of valuable lessons and improvement opportunities for the initiative. Furthermore, our analysis shows how the CVC framework can be used as a tool for self-assessment within a continuous improvement cycle, a methodological exercise which has a high potential for replicability in other multi-actor partnerships.

*3.2. Methodology*

A case study methodology is used in this investigation. A case study is typically used to investigate a contemporary phenomenon ("the case") in depth and within its real world context. Because they are based on a variety of data sources, case studies offer rich empirical descriptions of particular instances of a phenomenon [54]. The methodology has also been identified as useful in unveiling complex cause–effect relationships that offer lessons for addressing the major substantive themes in a field [55]. Over the last decades, case studies have been used extensively in multiple fields, including organizational theory [56], education [57,58], and strategy and decision science [59,60]. Their multi-disciplinary and cross-cutting nature thus makes them particularly suitable for exploring collaborative initiatives. Stott (2006) [61] particularly advocates the use of case studies in analyzing multi-actor partnerships.

The Alianza Shire is a unique case of a multi-actor partnership in terms of both the scope of the collaboration and the impact achieved since its creation in 2014. A wide variety of actors have been involved in designing and implementing innovative initiatives in order to provide energy access to refugees and host communities (see Section 4 for a detailed description of the composition and activities of the Alianza Shire). These actors are assisted in this collaboration by itdUPM, which assumes an intermediary role among them. The Alianza Shire thus provides analytical material of particular richness and interest for case study analysis and is well-suited to the research aims.

As well as their use for descriptive purposes, case studies can also play an important role in theory building and testing as they are guided by theoretical constructs that have multiple levels of analysis and a focus on specific processes, actors, or phenomena [62]. Our research, as outlined above, focuses on the transformational character of a multi-actor partnership as well as on the role of an intermediary actor in fostering that character using collaborative value creation (CVC) as its theoretical framework [27].

In order to simplify the different variables proposed by Austin and Seitanidi, and take into account the key organizational concerns encountered by the Alianza Shire's facilitating team, four categories were created to link together those with clear connections: organizational engagement, resources and activities, partnership dynamics, and impact (see Table 1).

**Table 1.** Categories used for analysis of the Alianza Shire.

| Categories | Original CVC framework variables |
|---|---|
| Organizational engagement | Level of engagement |
| | Importance to mission |
| Resources and activities | Type of resources |
| | Magnitude of resources |
| | Scope of activities |
| | Managerial complexity |
| Partnership dynamics | Interaction |
| | Trust |
| | Internal change |
| Impact | Co-creation of value |
| | Synergistic value |
| | Strategic value |
| | Innovation |
| | External system change |

Another important feature of the case study methodology is the use of multiple sources of evidence that can be triangulated in order to better substantiate findings [54]. Eisenhardt (1989) further advocates the use of multiple researchers in order to enhance the creative potential of a case study by combining different perspectives [62]. In this case, the paper was developed by a team of six researchers, three of whom are directly involved in the Alianza Shire and part of the itdUPM facilitating team that supports it.

The case study was conducted between 5 September and 28 October 2019 using the following sources of information: key documents associated with the different partnership activities (including project proposals, agreements, terms of references, contracts, internal regulations and norms), direct observation in the field (including attendance at meetings of the main governance bodies of the partnership: the Steering, Management and Communications Committees, and the Project Office, and open interviews with selected stakeholders). As this work focuses particularly on the intermediary role, those interviewed included current and former itdUPM members involved in the partnership in various capacities. Appendices A, B and C present brief descriptions of the different objects of analysis considered.

Analysis of the information gathered was conducted via an assessment of the partnership according to the different variables of the CVC spectrum. This was carried out independently by six individuals who have been involved in the facilitation of the Alianza Shire at different times. These diverse perspectives were subsequently discussed and contrasted in a two-hour structured group meeting which led to a preliminary global assessment of the partnership agreed upon by all those interviewed. The assessment enabled the composition of an initial narrative for the case which was then completed through several iterative work meetings among the authors of the paper. The final version of the paper was revised and validated by two key informants.

### 3.3. CVC framework as an Assessment Tool

In this paper, the CVC framework is used to conduct an assessment exercise within a continuous improvement cycle for the Alianza Shire. While other assessment tools and frameworks found in literature could have been used, we found that many of these are based on the assessment of goals linked to a project. While such an assessment is useful for estimating the effectiveness of specific activities in a given time frame, it does not evaluate the complex relationships between stakeholders in a partnership in depth, nor their evolution over time. Frameworks related to the usefulness of partnership as an approach can also be found [45], including collective–conflictual value co-creation [63], partnership outcomes assessment [64], partnership learning loop [65], partnership performance assessment [66] and the strategic scoping canvas [67]. Table 2 provides a summary of the main focus areas of each of them. While different aspects of these approaches were considered for this study, the CVC framework was selected for the exploration of the Alianza Shire because it stresses the value creation process among partners with an evolutionary perspective and a transformative aspiration; two approaches that have been highlighted in the literature on partnerships and which are particularly relevant to the SDG agenda [4,6].

**Table 2.** Frameworks for assessing multi-stakeholder partnerships.

| Partnership Analysis Framework or Tool | Main focus area |
|---|---|
| Collective-Conflictual Value Co-creation | Conflict between actors leads to innovation or general repositioning and impacts future value co-creation. |
| Partnership Outcomes Assessment | Improvements in partnership practice and exploration of a partnership's contribution to performance and outcomes. |
| Partnership Learning Loop | Focuses on reinforcement of collaboration between partners through different partnership layers. |
| Partnership Performance Assessment | Analyses partnership performance and effectiveness using three groupings of drivers: the external context, the organizational environment and the individuals representing each partner organization. |
| Strategic Scoping Canvas | Helps to clarify perspectives on strategic orientations and ambitions regarding the scope of the partnership and the depth of expected changes. |
| Collaborative Value Creation | Stresses the value creation process among partners with an evolutionary perspective and a transformative aspiration. |

## 4. The case of the Alianza Shire

The Alianza Shire was established in 2014 to develop innovate energy access solutions for those living in the refugee camps situated in the Shire region of northern Ethiopia where access to energy and lighting is limited and irregular. With a population numbering some 60,000 people in 2019, the refugees in these camps are mainly from Eritrea [68]. In order to improve the access and quality of energy services to this refugee population, two private companies and a corporate foundation in the energy and lighting sector—Iberdrola, Signify, and Acciona.org—joined forces with the Humanitarian Action Office of the Spanish Agency for International Development Cooperation (AECID) and the Innovation and Technology Centre at the Universidad Politécnica de Madrid (itdUPM) [69].

The diverse nature of the members is not accidental. Each of the three private sector representatives specializes in one or more areas of the energy and lighting sector, thus providing a broad and detailed knowledge base of the field. The roles of these partners are focused on their capacity to shape innovative solutions to energy access challenges through the involvement in the partnership of both selected specialists and executives. The AECID provides the institutional framework for the partnership, specialized knowledge in refugee protection, the support of the Spanish public sector and funds for its development. itdUPM, meanwhile, assumes a threefold role; firstly, as an academic institution, it provides the partnership with a large network of research experts and scholars; secondly, as a neutral forum, it assumes the role of partnership facilitator by curating the relations between members; and, thirdly, it works to ensure proactive project management and coordination.

Members of the Alianza Shire have consistently advocated that co-created innovative solutions should be implemented in projects that serve to enhance the quality of life of both host and refugee communities. Adopting a continuous improvement approach, solutions are co-designed together with relevant stakeholders in the field—including final users—and, once implemented, their performance is monitored and improved.

The Alianza Shire partners acknowledged the importance of developing projects that could be scaled-up by starting with interventions of a reduced dimension and a greater likelihood of success from the start. Between 2014 and 2017, a first pilot project was developed in one refugee camp. This prototype was aimed at demonstrating the potential of working in partnership to address energy

challenges while also serving as a space for testing and improving innovative solutions and work methodologies.

The positive outcomes arising from the pilot project indicated that the partnership had the potential to address complex problems. Findings were shared and discussed with representatives from international stakeholders such as the United Nations High Commissioner for Refugees (UNHCR), the European Union, and various energy access networks and initiatives, including the Moving Energy Initiative and Safe Access to Fuel and Energy [70]. Both the project results and positive feedback from these international organizations prompted members of the Alianza Shire to take the decision to scale up. The second phase of the project, co-funded by the European Union's Emergency Trust Fund for Africa [71], started in 2018 and will be implemented by 2021. The impacts of both the prototype and Phase II of the project can be consulted and compared in Table 3.

The Alianza Shire's work in Ethiopia has been accompanied by and coordinated with UNHCR which is invited to all the partnership's Steering Committee meetings. Other key partners in the field include the Norwegian Refugee Council (the Alianza Shire's implementing partner in the pilot project), ZOA (implementing partner in Phase II), and the Ethiopian Agency for Refugee and Returnee Affairs (ARRA) which is responsible for managing refugee camps in Ethiopia in collaboration with UNHCR. The involvement of these partners ensures the direct participation of refugee communities in the design and implementation of energy access solutions in the Shire camps.

**Table 3.** Scaling up the impact: from prototype to project.

| Pilot project: The prototype | Phase II: The project |
| --- | --- |
| 1 refugee camp | 4 refugee camps and host communities |
| 8000 refugees | 60,000 refugees<br>17,000 people from the host community |
| 4 km of street lighting | 25 km of street lighting<br>100% connection of communal services<br>Connection of 450 private businesses |
| Training of refugees | Managerial training for the Ethiopian Electric Utility's regional managers<br>Technical training for the Ethiopian Electric Utility's field staff<br>Training of trainers for implementing partner staff<br>Training of refugees and host community |
| Trained refugees working for NGO in the camps | Trained refugees and host community included in Ethiopian Electric Utility structure |
| Only on grid component (street lighting and electrical grid improvement) | On grid in 4 camps + 1700 third generation solar home systems (SHS)<br>Creation of 6 micro-businesses owned by refugees and host community – in charge of the operation and maintenance of the SHS |
| Total budget of 500,000 € (funds and in-kind contributions) | Total budged of 4,700,000 €<br>(funds and in-kind contributions) |

In order to align with international agreements and initiatives, the Alianza Shire works within the 2016 United Nations Comprehensive Refugee Response framework (CRRF), the implementation of which is being piloted in Ethiopia [12]. While seeking to design and implement the Phase II project in line with the mission and vision of the CRRF, the Alianza Shire also intends to share results and lessons learnt through the use of a partnership approach with the international community.

Managing a partnership of this magnitude is not an easy task. It requires careful consideration of the different backgrounds, organizational cultures and objectives of the five different organizations that are working together, as well as those of local implementing partners and collaborating

organizations. Because of this, a comprehensive managerial and governance structure has been designed. Based on the logic of continuous improvement, this structure has evolved over the five-year lifetime of the partnership.

Because the complex governance structure combines diverse professionals, it requires a strong degree of formalization to ensure that each organization and actor is provided with a clear mandate and that the relationships between them are fluid. The relationship between the internal stakeholders (partners) and other external entities and bodies of the Alianza Shire is thus formalized through a number of protocols and guidelines, some of which are binding. Details of the different management structures and their relationships are provided in Appendix B.

Today, due to the dedicated engagement of the five core members and the trust that has grown between them, the Alianza Shire has reached a stage of maturity. A space has been developed in which organizations with very different working cultures and backgrounds are able to collaborate in order to find innovative solutions to the complex problem of improving energy services for people who are forced to flee from their home countries and temporarily settle in refugee camps. Reaching this level of trust has taken four years and required a huge effort on the part of the facilitating entity as well as an important personal commitment from key representatives of the member organizations. The different stages of the Alianza Shire's development can be seen in Figure 2.

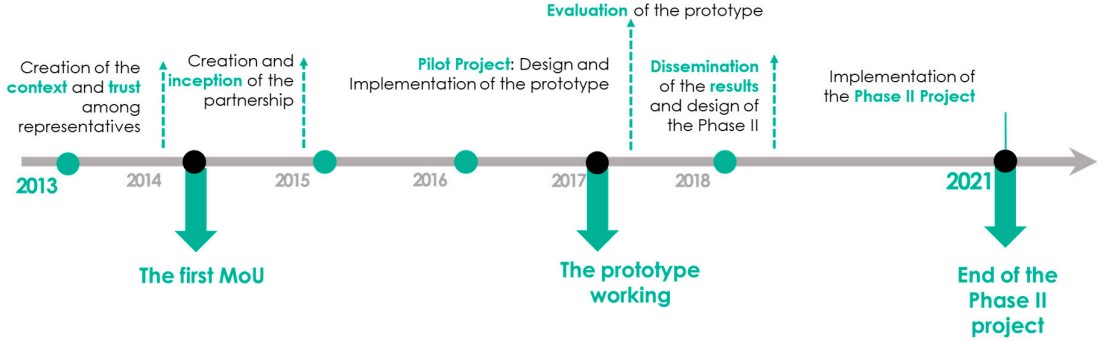

**Figure 2.** Stages of the Alianza Shire.

## 5. Results: CVC analysis of the Alianza Shire

In this section, an analysis of the Alianza Shire is presented in relation to the CVC framework. As outlined in Section 3 (the research approach), four different dimensions have been considered: *organizational engagement, resources and activities, partnership dynamics, and impact*. Each of these dimensions is explored in terms of the sources and types of value created using the terminology of the CVC spectrum. This has enabled an assessment of the transformational character of the partnership in terms of its position within the CVC continuum. A synthesis of this analysis is provided in Table 4.

**Table 4.** Analysis of the Alianza Shire using the CVC framework.

| Nature of relationship (CVC framework) | | Status at start (2014) | Current status (2019) |
|---|---|---|---|
| **Organizational engagement** | Level of engagement | Moderate | High |
| | Importance to mission | Peripheral | Central |
| Resources and activities | Type of resources | Core competences | Core competences |
| | Magnitude of resources | Small | Big |
| | Scope of activities | Narrow | Broad |
| | Managerial complexity | Simple | Complex |
| Partnership dynamics | Interaction level | Low | Intensive |
| | Trust | Modest | Medium |
| | Internal change | Minimal | Medium |

| | | | |
|---|---|---|---|
| | Co-creation of value | Medium | High |
| | Synergistic value | Occasional | Predominant |
| Impact | Strategic value | Medium | Major |
| | Innovation | Medium | Frequent |
| | External system change | Small | Significant |

The intensity of the color in the last two columns represents the extent to which the element under consideration is transformational. The scale of analysis is that proposed by Austin and Seitanidi (2012).

## 5.1. Organizational Engagement

Although the Alianza Shire was formed in 2014, the organizational commitment of the different participating organizations at that stage was limited. While involvement in the partnership was clearly of potential interest to them, there was no clear rationale and framework for their participation in this type of initiative. Indeed, although there were other multi-stakeholder partnerships operating in the area of humanitarian aid at the international level, in Spain there were no precedents for this [72]. The AECID Master Plan at that time (2013–2016) did not explicitly include multi-stakeholder partnerships as a tool for working in international cooperation and development. Furthermore, there was no clear formulation of the extent to which the partnership could be *linked to the interests* of each member organization. When the Alianza Shire was created, the *importance of the partnership to the mission* of the participating organizations was perceived as peripheral and the *level of engagement* of the partners could be characterized as moderate.

The SDGs, and in particular SDG 17, have been important drivers for change in this regard as they have provided all partners with a clear and shared framework that positions multi-stakeholder partnerships as crucial for achieving the goals of the 2030 Agenda. It can be argued that multi-stakeholder partnerships in general and the Alianza Shire in particular have since become *central to the mission* of the different partners. This is illustrated by the different statements and organizational documents from the Alianza Shire partners, provided in Appendix C.

Participation in the Alianza Shire is also linked to the social responsibility agenda of the private sector organizations involved. In Iberdrola, for example, the partnership is viewed as one of the most important initiatives in the company's "Electricity for all" program, one of the cornerstones of its social responsibility strategy [73]. The achievements of the Alianza Shire are also consistently mentioned in the annual sustainability reports of the three private-sector members [74–76]. In addition, the Alianza Shire appears in the progress report of the Government of Spain for the implementation of the 2030 Agenda [77].

The SDGs have become a core strategic element of itdUPM's work and central to a number of flagship activities that promote the 2030 Agenda among different target populations. Through a series of bottom-up co-creation seminars for faculty and researchers, itdUPM has contributed to the elaboration of a new SDG-aligned research strategy for UPM [78]. The Alianza Shire is a central part of this strategy and seen as an important vehicle for demonstrating the potential of partnership for the SDGs, both within and outside the University. The partnership illustrates two core principles of itdUPM's theory of change: the transformational power of multi-stakeholder partnerships and SDG 17, and the importance of prototype scaling up in order to achieve wider impact.

While working towards the achievement of the SDGs, the Alianza Shire has grown considerably in terms of scope, impact and public visibility. In consequence, the associational value in terms of reputation and credibility of its members has increased. Interdependence and collective action have become the modus operandi of the partnership and the organizational engagement of the different partners, which has increased exponentially, can now be assessed as high. Although there have been significant changes in the organization and structure of some of the partner organizations, newer members of the Alianza Shire's Steering Committee have continued to support the initiative with the same enthusiasm as their predecessors. Testimonies from the top management of the different partner organizations (CEOs, Secretary of State, Rector of UPM) that highlight the achievements of the Alianza Shire can be found in Appendix B.

The growth of levels of commitment and the importance of the Alianza Shire for the mission of its partner organizations has relied to a great extent upon the facilitation role played by itdUPM in (a) generating a common language and narrative for the partnership and seeking to align the objectives and incentives of all the partners, (b) working to promote complementarity between the partners through activities that highlight the importance of the SDGs for them, and (c) highlighting SDG 17 as an essential mechanism for achievement of the SDGs and the Alianza Shire as an example that demonstrates what can be accomplished by working in partnership. In collaboration with Iberdrola and the Acciona Foundation, itdUPM has also developed University–Business chairs on the theme of the SDGs and access to energy, and has contributed to the elaboration of the AECID's current Master Plan.

## 5.2. Resources and Activities

Since the partnership began, its members have acknowledged the importance of designing projects with a scale-up perspective. The initial activities that were developed and implemented by the Alianza Shire were narrow in scope and the magnitude of resources was small. Since then both the input of resources and scope of activities have been considerably scaled-up for the Phase II project (see below).

In addition to the operational objectives (safe energy supply to 8000 refugees), the pilot project set an organizational objective: to serve as a test bed to develop and validate an innovative partnership approach with efforts directed towards creating an enabling space for this. In line with the CVC framework, an attempt was made to develop interaction value, i.e., shared language, knowledge, joint problem solving, common technical and organizational approaches, communication and coordination. Regarding the type of resources put into play, from the very beginning of the partnership, member organizations agreed that projects would not be based on monetary exchange (in fact the private companies involved have not made a single direct financial contribution—see the Alianza Shire MoUs in Appendix A) but in the creation of synergistic value based upon using each partner's core competences.

Following the experience of the pilot project, the scope of activities has expanded considerably (see Table 3). In Phase II, the Alianza Shire has applied a more diversified range of technological approaches and started to develop non-technical activities aimed at strengthening refugee–host community integration, generating local economic activity and employment, and building the capacity of key stakeholders such as the Ethiopian Electric Utility (EEU). The idea is to develop a comprehensive approach with the development of solutions adapted to the partnership's specific social and regulatory context and to ensure that these solutions are sustainable in the long run. At the same time as seeking to achieve local transformation, the Alianza Shire hopes to contribute to change at a more global and systemic scale. This has led to an increase in activities aimed at systematizing knowledge (technical and organizational approaches and lessons learnt) and sharing this knowledge with the international community.

Developments in the scope and scale of activities have required a significant increase in the magnitude of resources. The amount of economic resources allocated to the Alianza Shire's projects has increased tenfold (see Table 3). While the pilot project was funded with a grant from the AECID and the contributions of partners, Phase II is co-funded by the European Union. In terms of human resources, the pilot project relied on the experience and knowledge of a few experts from each organization. In Phase II, these resources have been expanded and diversified as the organizations have assigned specialists for specific project tasks in order to enhance impact. This has resulted in a combination of different profiles that few humanitarian organizations are able to gather together on their own, including business model specialists, diplomats, scientists, lawyers, experts in safety and security, among others. The interaction among these specialists is rich and frequent and has resulted in mutual learning and the generation of benefits in terms of transferred resource value. Many of these specialists have a permanent or part-time dedication to the Alianza Shire which adds to their professional portfolio. Some technicians from private companies participate in the activities of the partnership on a voluntary basis. While this has had a positive impact on corporate volunteering

programs, it also raises management challenges in terms of the lack of time available for this input during specific periods of activity and the complexity of matching field missions to holiday periods.

The expansion of the Alianza Shire in terms of scale has been driven by an evolutionary management structure, the complexity of which has steadily increased. At its inception, the partnership comprised two committees and a group of experts who operated without systematized procedures. Currently, the partnership is governed by five different bodies and a number of work plans, procedures, and protocols are in place. In addition to the staff of partner organizations, almost 100 people are involved in the partnership. In the first year of Phase II (2018–2019), much effort has been devoted to designing and implementing a set of agreements among some of the twenty or so organizations that are directly involved with the partnership. These agreements regulate both relationships and commitments between organizations as well as project management procedures and processes.

The managerial structure of the Alianza Shire has adopted a traditional project-oriented approach based on internationally recognized guidelines [79]. However, this conventional structure has been complemented by additional processes and methodologies aimed at driving the transformational momentum outlined in Kotter's dual operating system approach [80]. These seek to provide an independent space in which all partners reflect on how to foster innovation and manage knowledge.

The facilitator plays an important role in terms of enabling the increase in the scale and range of the resources and activities of the Alianza Shire. itdUPM acts as an official grant recipient and takes care of the administrative and accountability obligations that this implies. It also works closely with each organization to find the best strategies to address internal resistance and to build internal capacity by leveraging the competencies and complementarities of each partner. Furthermore, itdUPM has fostered specific initiatives to reinforce the capacity of the partnership by, for example, promoting the creation of an interdisciplinary group of more than 15 UPM researchers working as experts in fields such as agriculture, construction, ICT, and water management. This group has developed a technological needs assessment methodology that adopts an interdisciplinary approach [81]. The results of the application of this methodology will be used by the Alianza Shire and its implementing partners in the field for the design of future actions. itdUPM has also put in place a continuous improvement process which has enabled the identification of opportunities for enhancing the overall design and management process, and greater participation of partner organizations. Finally, as a partnership facilitator, itdUPM works to systemize and disseminate the knowledge derived from project activities.

### 5.3. Partnership Dynamics

The interaction level of the Alianza Shire partners has increased since its creation in 2014. During the first phase of the Alianza Shire, the Steering Committee, which met on an ad hoc basis, was the only formal body for partner interaction. There are now four established partnership committees that function and interact systematically. Participation in these bodies has engendered considerable interaction value in terms of coordination, transparency, and joint problem-solving.

Beyond the interactions within the formal governing bodies, the partners have also become increasingly involved in the diagnosis of needs and the co-creation of appropriate solutions for energy access challenges. This co-creation of value has also included the refugees and their host communities. By increasing co-creation and interaction, complementary skills and knowledge have been identified, which have increased the potential for resource complementarity and synergistic value. Furthermore, the Alianza Shire serves as a valuable experimentation space for partners to test different methodologies and technologies and has the potential to enhance their capacity for innovation. The level of trust among members of the partnership has also increased. Constant interaction throughout the six years of the partnership's life, as well as the results and the impact achieved, has generated trust, first among individual partner representatives and then between organizations.

The development of the Alianza Shire into a large-scale project has also posed difficulties. Growth in scale has increased levels of pressure on partners in terms of reputational risks. This is particularly so for itdUPM which has dual accountability to both the EU (the Phase II project funder) and the AECID (the grant recipient) for management of the funds awarded and ensuring expected results. In addition, as the activities of the Alianza Shire have expanded, the participation of individuals and departments of member organizations that had no previous contact with the partnership has increased. This is the case, for example, for human resources, security, and legal departments for whom the context of the partnership's operational setting is a challenge and adaptation costs are generally high.

Partnership dynamics are also challenged by insufficiently adapted regulatory and administrative frameworks. At the European level, funding schemes are designed with a conventional project-oriented logic. Project-oriented mechanisms to guarantee the commitments that an organization acquires from a third party are not designed to facilitate long-term and flexible collaboration among organizations. Monitoring and accountability are generally based on a control and hierarchy rationale rather than on mutual trust and decentralization. As a result, relationships among partners tend to follow conventional donor–recipient or transactional patterns.

This disconnect is also evident in the Spanish policy context where there is no regulation for multi-actor partnerships with long-term transformational goals. Furthermore, the internal policies and procedures of the different partners in key aspects such as security management are not aligned. As a result, the formalization of agreements among members is frequently a long and exhausting process.

Both the increase in pressure and regulatory challenges are sources of tension in the partnership and pose continual barriers to the increase of mutual trust among members. Against this background, the partnership facilitator is a key enabling actor. As well as taking on most of the bureaucratic burden resulting from internal and external misalignment, itdUPM has also assumed responsibility for the formalization of agreements and acts as a mediator among members for this purpose. In undertaking this work, itdUPM leverages its identity as a university center that is seen as a neutral and trustworthy player able to build positive connections between organizations. Universities can also easily develop spaces for dialogue and exchange. An example of this is the Master in Strategies and Technologies for Development, a postgraduate level program managed by itdUPM [82] that serves as a meeting place for some of the individuals and organizations that make the Alianza Shire possible. Key partner representatives are invited as guest professors and Master's students also take up internships in partner organizations. Through their interaction within the Master's program, the different organizations can thus reinforce relationships that generate mutual trust.

*5.4. Impact*

As explained in the section on organizational engagement (Section 5.1), the *strategic value* of the partnership has increased since its creation. This increase in strategic value can be attributed to the promotion of the 2030 Agenda and the increasing importance of the SDGs and multi-actor partnerships for partner missions. At the same time, the increase in scale of partnership activities has multiplied the value of the Alianza Shire for partner organizations.

*Co-creation* and the idea of generating value through a participatory process has always been an essential component of the Alianza Shire. Participation of all key stakeholders in the conception and implementation of solutions is a key pillar of the Alianza Shire's theory of change. Indeed, for the Alianza Shire, broad and inclusive participation is fundamental for the creation of solutions that are truly transformational and sustainable in the long term. Co-creation and participation have steadily increased since 2014 both in terms of governance and with regard to activities in the field. The outcomes of the pilot project, which provided access to safe energy and lighting for 8000 refugees, were encouraging. This contributed to a 60% reduction in incidents such as night burglaries, a reduction in the collection of firewood for cooking of 1500 tons per year, an annual reduction of 2000 tons in $CO_2$ emissions, and an annual economic saving of 30,000€ in diesel for camp operators [83].

With regard to innovation, an important goal of the Alianza Shire has been to demonstrate that it is possible for non-traditional actors in the humanitarian field to work together comprehensively with the Spanish Humanitarian Action Office in order to offer energy access solutions in refugee camps. Great emphasis has also been placed on the future sustainability of pilot actions. In Phase II, many more innovation processes are being deployed. These are summarized below using Tidd and Bessant's "4Ps of innovation" [84,85].

- Product—changes in products or services: High quality products that are framed holistically such as use of third generation solar home systems (SHS), robust and adapted LED street lighting technology, high quality electrical grid materials, and augmented reality equipment for training.
- Process—changes in the ways services are created or delivered: Development of a management model based on guaranteeing sustainability through diverse contributions and coordination among different actors. Six micro-enterprises (between refugees and people from host communities) with their associated business models will be created to ensure sustainability in the management and maintenance of the 1700 SHS.
- Position—changes in the way services are presented to the user and how these are communicated and reframed by government and other actors: Positioning energy supply as a central element in the management of the Shire refugee camps and as a catalyst for new development possibilities. This has been possible thanks to the integration of two key partners in the field and co-creation efforts with them: ARRA (Ethiopian Agency for Refugee and Returnee Affairs, responsible for managing refugee camps in collaboration with UNHCR) and the Ethiopian Electric Utility.
- Paradigm—changes in the underlying mental models that shape what the service offers: Contributing to a shift in the traditional humanitarian response mindset by presenting refugee camps as a source of innovation where refugee and local communities can participate in the design and implementation of solutions to the challenges they face. This also suggests that non-traditional actors such as academic and private companies can play a clear and positive role in the humanitarian field.

The innovation capacity of the Alianza Shire has underpinned the sustainability of the Phase II project, which is strongly aligned with the sustainability drivers of access to basic services programs and specifically on off-grid energy access projects [21,86–89]: technology quality assurance, assessment of community payment capacity, local training and awareness campaigns, operation and maintenance, supply through business models, stakeholder and agreements management, and influence on public policies.

When the Alianza Shire began, external system change was very limited as attention was placed upon creating the "right" conditions for the partnership to operate. However, due to the positive results of the pilot project and the work that is being developed in Phase II, the possibilities for influencing external key actors and wider forums are growing and current external system change can be considered as significant. In the context of the Spanish private sector, considerable efforts are being invested in encouraging practical inter-sectoral collaboration and dialogue. The "SDG17 GoODS Award", which is granted by the Spanish branch of the UN Global Compact was awarded to the Alianza Shire in 2019. In the academic sector, itdUPM is sharing the Alianza Shire as an example of how to connect the university with social innovation spaces, an idea that was given recognition with the "Best Paper Award" at the 2017 International Conference on Sustainable Development held at Columbia University in New York. The Alianza Shire is also attracting the attention of the international humanitarian community. The partnership's ability to connect private sector core competencies with humanitarian innovation processes driven by the demands of key actors was acknowledged with the Set4Food Humanitarian Energy Award which was presented at the First Humanitarian Energy Conference [90]. As a result, the Alianza Shire will be one of the fifteen practical initiatives presented at the First Global Refugee Forum Marketplace of good practices [91].

There is a clear intention on the part of itdUPM to systematize and disseminate all these processes. However, since Phase II is such a complex project, there is some concern that the operational needs of the project will make this task difficult. To address this, itdUPM, in its role as partnership facilitator, is designing a series of knowledge management activities that will be integrated into a program that is complementary to the project.

*5.5. Assessment of the Evolution of the Transformational Character of the Alianza Shire*

A key finding of the global assessment of the Alianza Shire in relation to the CVC continuum is that the Alianza Shire was not conceived as a classical philanthropic partnership among different stakeholders (Stage I of the continuum). With a focus on establishing an innovative initiative of strategic importance for the different partners in which co-creation played a central role as a fundamental principle at all levels, the transformational ambition of the partnership was apparent from the start. Although the scale of operations was modest at the beginning, the Alianza Shire was also created with an iterative and scale-up logic. As a result, while the Alianza Shire can be placed at the CVC-*Stage I* position in terms of scale (in relation to variables such as the magnitude of resources or the scope of activities) it falls within the *CVC-Stage II* or even *CVC-Stage III* categories for other variables (i.e., level of engagement, type of resources, co-creation value). In Phase II, the scale of the Alianza Shire has dramatically increased in transformational potential and it may consequently be placed between Stages III to IV of the CVC continuum. This analysis has further enabled the detection of a series of barriers that may impede the Alianza Shire from realizing its full transformational potential. These barriers will be analyzed in the following section.

## 6. Discussion

*6.1. Results Discussion*

With regard to content findings, a general conclusion that can be made is that the Alianza Shire has not evolved through the classic stages of the CVC collaboration continuum (philanthropic, transactional, integrative, and transformational); rather it was created with a clear transformational aspiration with partners exchanging resources related to their core competences in a shared manner and co-creation as an important operating principle from the start. However, the idea that collaborative initiatives follow an evolutionary path is still valid for our case study. Indeed, the Alianza Shire was established with a progression perspective in mind: it started with a prototype and the investment of a modest amount of resources and developed to become a partnership that is now significant in terms of scale. The variables that have been considered in relation to the scale of operations (i.e., those linked to resources and activities) have consequently experimented a progression which can be conceptualized through the CVC continuum. Partnership dynamics have also experienced a remarkable evolution as, in spite of a significant initial level of engagement, the current positive levels of trust and interaction among the member organizations have been built through continuous effort over time. In conclusion, although our case study reflects the evolutionary nature of collaborative initiatives, it also suggests that multiple paths are possible for this evolution and that an ex-ante aspiration for transformation and a scale-up strategy may be relevant for progression to a transformational stage.

Through the CVC framework analysis, the importance and relevance of the facilitator for this partnership has been revealed. By promoting the creation of synergistic value, the facilitator role has strengthened the innovation capacity of the Alianza Shire, underpinning the sustainability [21] of its operational activities. By acting as an "enabler-connector" for partnership dynamics, the facilitator has overcome barriers to collaboration between heterogeneous profiles such as those of refugees, Ethiopian Electricity Utility technicians, private sector energy experts, diplomats, and academics, stimulating inclusive planning and designing approaches [16] and market based strategies [17,19].

The authors note the usefulness of incorporating the intermediary role within the CVC spectrum and continuum in order to improve its use as an assessment tool. The evolutionary vision that the

CVC proposes and the categories of analysis have been helpful for appraising the facilitator role in the Alianza Shire in a manner that is easily transferable to other partnerships and intermediaries. The CVC framework has also enabled the authors to conduct a thorough assessment of the partnership and identify a number of barriers to transformation, as well as possible mitigation measures. Two key conclusions emerge: (1) The main barriers detected are primarily related to "partnership dynamics", most particularly to difficulties in developing trust and fostering the internal changes needed to work within a multi-stakeholder collaboration setting; and (2) these barriers arise at different levels and relate to individuals representing partner organizations, the partner organizations themselves, and the wider external environment. The barriers identified, which are consistent with those in partnership assessments using other evaluation frameworks [64,66], are outlined below:

- Individuals: the high adaptation costs of integrating individuals from different partner organizations as actors in key partnership processes (e.g., legal, security and safety, procurement) hinder interaction. Some roles in particular are not fully understood during the initial stages of interaction. For example, the partnership facilitator is commonly confused with the Project Office, the partner organizations are occasionally considered to be external stakeholders, and competitors and local partners in the field are at times viewed as service or product suppliers.
- Partner organizations: Internal consolidation of the partnership is a long-term process. The silos that exist in all organizations, especially large international enterprises or public bodies such as those participating in the Alianza Shire, limit the involvement of different departments and business units and thus reduce the potential for co-creation within and between organizations.
- External environment: Formal management, monitoring, and accountability mechanisms are not adapted for working in a long-term partnership with a transformational aspiration. This may undermine organizational confidence in processes involving fund allocation, sharing of responsibility, and the participation of external actors.

In order to address these barriers, the Alianza Shire case study shows that a partnership facilitation function is essential. This study has assisted in promoting a clearer understanding of the core role and mission of a partnership facilitator: to support a partnership to evolve towards a transformational stage. The article further contributes to calls for more sharing of practical cases that support deeper learning about the role of facilitators in transformative partnerships [4,5,46,92,93]. The partnership facilitator can fulfill this role in the following ways:

- Generation of a collaboration context: Promoting organizational engagement by encouraging partners from different sectors to assume their primary role and generating trust through a deeper understanding of the identities and views of different parties [5].
- Design: Promoting the generation of shared value with co-creation of activities among partners and facilitation of a framework for systematic management, coordination, and continuous improvement.
- Mediation: Facilitating key interaction processes, creating a neutral space for dialogue and addressing procedures that are not adapted for long-term transformational partnerships.
- Promotion of key transversal processes such as innovation, learning, and gaining wider influence.

*6.2. Limitations of the Study*

A clear limitation of this study is the fact that the conclusions are derived from a single case study. Although the Alianza Shire provides complex and rich case study material, further studies in this field are needed, including different collaboration and intermediary settings and contexts.

The authors have focused in this article on the pillars of the CVC framework described by Austin and Seitanidi in 2012 [26]: Spectrum and Stages. Austin and Seitanidi have subsequently enriched the framework with other dimensions: Mindset, Collaboration Process, and Outcomes [86,87]. A further in-depth study should address these complementary dimensions.

In conducting the assessment of the Alianza Shire, the authors have adopted a qualitative approach. Although significant practical and methodological implications can be derived from this analysis, in order to enrich the appraisal in the future, it would be useful to develop a quantitative measurement scale for the different components of the CVC framework.

### 6.3. Future Directions of the Research

The practical application of the CVC framework presented several weaknesses that hinder a more comprehensive partnership analysis. To address this, some suggestions for future developments are presented below:

- In order to simplify the analysis and avoid too many interactions between elements, it is helpful to group the wide variety of elements of the CVC framework into categories. This paper presents a proposal for grouping these elements into "organizational engagement", "resources interaction", "partnership dynamics", and "impact" but other groupings may also be appropriate.
- Increase the granularity of the analysis so that the interaction of a number of common elements within each category can be analyzed; for example, the partnership's project portfolio, its ecosystem of people and organizations, and the tools and methodologies used.
- Incorporate analysis of the wider context which the project aims to influence, including political changes and regulatory frameworks. These elements could be included in the category of "partnership dynamics".

During the preparation of this article, a range of tools and frameworks have been revealed as highly useful for analysis of the Alianza Shire. To consolidate the future stability of the partnership, a tool that may help align visions among the partners is the strategic scoping canvas [67]. Its use will be proposed in the ongoing reflection and learning processes of the Alianza Shire. In order to increase diversity and inclusiveness in the design of future actions [16], an "out-of the-box thinking" proposal has also emerged from the CVC-related research [94] for working to better connect and create value with different partner "end users" such as company clients, university students, and refugees.

With regard to the intermediary role, it is unlikely that the features and characteristics that we might expect of partnership facilitators will reside in a single individual. Instead, the Alianza Shire case study suggests that this kind of facilitation work should be undertaken by an organization that incorporates a team of individuals. In this case, the role has been undertaken by a university centre able to offer support to the Alianza Shire because of its perceived neutrality, commitment to the transformational agenda of the SDGs, and the creation of safe spaces for co-creation.

The intermediary role in partnerships is one that clearly merits deeper research. In order to explore this more fully, interactions between the CVC approach and the sustainability transitions theory (see above, Section 2) merit attention as intermediaries have received increased attention within this strand of research [39,95,96].

### 6.3. Concluding Remarks

As discussed throughout this article, new forms of collaboration between people and organizations are essential for the achievement of the 2030 Agenda. The urgency and complexity of the problems faced and the changes required to address them clearly demand responses that are transformative in nature [3,4]. For this reason, as Austin and Seitanidi (2012) have suggested [27], it is pertinent to ask what key enablers can assist partnerships in reaching a transformational state. The experience of the Alianza Shire suggests that two important factors are critical for supporting this: a facilitating organization that ensures the creation of shared value, and partners that possess both the aspiration and a cohesive strategy for working together to achieve significant systemic change.

**Author Contributions:** Conceptualization, J.M.S; methodology, J.M.S., T.S.C.; validation, T.S.C, L.S.; formal analysis, J.M.S., A.A.; investigation, J.M.S., J.M.A.; writing—original draft preparation, J.M.S, T.S.C, A.A, L.S.; writing—review and editing, J.M.S., L.S., J.M.A, C.M.; visualization, J.M.S., A.A.; supervision, T.S.C., J.M.A., C.M.; project administration, J.M.A., A.A.; funding acquisition, C.M., J.M.A. All authors have read and agreed to the published version of the manuscript.

**Funding:** This doctoral research has been supported by funding from Universidad Politécnica de Madrid (UPM) and Iberdrola-UPM Chair on Sustainable Development Goals. The Alianza Shire Phase II Project is funded by the European Union and the Spanish Cooperation (AECID); and cofounded by in-kind contribution of UPM, Iberdrola and Acciona Foundation.

**Acknowledgments:** The authors would like to acknowledge the hard work of the people conforming the different bodies of the Alianza Shire (Steering, Management and Communications Committees, and Project Office), and the engagement of refugee people and practitioners working in the Alianza Shire's actions on the field. The authors would also like to acknowledge the indispensable contribution of the Alianza Shire's facilitating team that are not signing this paper: Ruth Carrasco and Alejandra Rojo (former Alianza Shire Coordinators), Manuel Pastor (Operations Officer), Xose Ramil (Communications Officer) and Dalia Mendoza (Innovation Assistant).

**Conflicts of Interest:** The authors declare no conflict of interest. The funders had no role in the design of the study; in the collection, analyses, or interpretation of data; in the writing of the manuscript, or in the decision to publish the results.

## Appendix A

This appendix contains additional information on some of the partnership's core documents and agreements. Due to the confidential and sensitive nature of their content and the agreements contained therein, the information includes an overall description rather than full details. Further information on any of these documents may be requested from the authors of this article.

**Table A1.** Alianza Shire: Core documents and agreements.

| Element | Description | Additional Considerations |
|---|---|---|
| Alianza Shire Memorandum of Understanding (Pilot Project) | Agreement for the creation, operation and evaluation of the Public-Private Humanitarian Action Partnership for the development of the Pilot Project. The purpose of the Agreement is to establish the necessary working mechanisms for the development of the Pilot Project, through the creation of a multi-stakeholder humanitarian action partnership between the members. | Signed by the 5 Alianza Shire members. The members agree to share the common objective of developing innovative and sustainable solutions that consider the needs and aspirations of the designated population. |
| Alianza Shire members Agreement (Phase II) | Agreement for the creation, operation and evaluation of the Public-Private Humanitarian Action Partnership for the development of the Phase II Project. The purpose of the Agreement is to establish the necessary working mechanisms for the development of the Phase II Project through the creation of a multi-stakeholder humanitarian action partnership between the members. The Agreement includes a section that gathers the overall governing | Signed by the 5 members of the Alianza Shire. Contents and wording finalized (process of almost a year's duration), pending authorization from public administration. The members agree to share the common objective of developing innovative and sustainable solutions that consider the needs and aspirations of the designated population. The Agreement includes aspects such as the governing structures and member representatives in each Committee, commitments and economic contributions; code |

| | processes and principles of the partnership that have been further developed together with the Steering Committee. | of conduct; security management and multiple administrative, legal and binding aspects. |

**Table A1.** *Cont*

| Element | Description | Additional Considerations |
| --- | --- | --- |
| Project Management Plan (PMP) | Intended to be an operational guide for the integrated management of the Project. It is a tool at the disposal of the different partnership organs and seeks to organize their work. It establishes the functions and responsibilities of each body and the different project management processes and procedures. It is a tool that facilitates the execution of the project in all its stages. | The PMP aims to be useful, easy to use, agreed among all parties and subject to continuous improvement.<br>It follows international standards such as the PMBOK (Project Management Body of Knowledge) and ISO 9001:2015<br>It gathers, among other elements, the scope, cost, time, risk, quality and communication management protocols, as well as the organizational structure and main innovation, execution and coordination processes.<br>The PMP includes, among others, key processes such as internal training, knowledge management, strategic evaluation, seminars and community participation processes. |
| Security Agreement | The purpose of this agreement is to govern the terms and conditions applicable to the collaboration of the members and ZOA in order to ensure the security of the Project and the personnel travelling to it during their time in the Tigray Region. It includes both ZOA and member obligations, security incident management processes and different legal and administrative aspects. | The process has been ongoing for well over a year. It has involved the participation of the legal and security departments of all members.<br>Signing was accelerated (and made feasible) due to the personal commitment of the former ZOA Company Director in Ethiopia.<br>It has involved the creation of an ad-hoc evacuation plan for the Alianza Shire, an internal emergency situations management protocol, the creation of the Alianza Shire Emergency Committee and an addendum to the agreement. |
| Communications Protocols | The communications protocols govern the Alianza Shire external communication and visibility Procedures. They include the Communication and Visibility Plan, the Alianza Shire Key Messages document and communication guidelines for partners in the field. | All the protocols have been produced by itdUPM, some of which have been revised by members and approved by the Communication Committee.<br>The Communication and Visibility Plan was rejected by the Steering Committee on three occasions before its definitive approval.<br>Interest has specially been focused on the visibility of each member in the communication actions and materials. |
| Agreements with partners in the field | These agreements seek to govern the relationship between local partners in the field and the Alianza Shire with regard to the implementation of the project.<br>Some agreements – Memorandums and Letters of Understandings (MoUs and LoUs) – are non-binding documents in which the organizations mutually recognize | The MoUs and LoUs reflect willingness but do not ensure specific collaboration or support. The Grants are based on traditional cooperation schemes whereby an organization assumes the responsibility of executing a series of activities and achieves certain goals.<br>The Grants and, to a lesser extent, the MoUs and LoUs, may be seen as contractual agreements for service exchange in which there is little room for transformation or innovation. |

each other and reflect their intention to collaborate.

Others – Grants – are binding documents through which a partner agrees to work in the project, assume specific responsibilities, and obtain an economic contribution.

## Appendix B

**Table A2.** Additional information and supporting evidence regarding essential elements of the Alianza Shire.

| Element | Description / Key Indicators | Additional Considerations | Supporting evidence |
|---|---|---|---|
| Pilot Project | Training for refugees and host community (practical part directly related to grid extension works)<br>Creation of group of operators under Norwegian Refugee Council<br>4 Km of street lighting<br>All communal services connected to the grid<br>1 refugee camp (8000 people) | Positive impact (according to preliminary assessment)<br>Demonstration of partnership-based approach and driver for Phase II<br>Solutions implemented have not proven to be sustainable (Operator team and overall grid situation)<br>Several lessons learnt for consideration in Phase II | Alianza Shire project description<br>Alianza Shire case study<br>Appearances in mass-media<br>Pilot Project Technical Report |
| Phase II Project | Training for Ethiopian Electric Utility (EEU) staff and Training of Trainers approach<br>Training for refugees and host community (practical part directly related to grid works)<br>Labor insertion of operators in EEU<br>Connection of 450 private businesses in the camps<br>+25 Km of street lighting<br>All communal services connected to grid<br>4 refugee camps and host communities (+40, 000 people)<br>Training for refugees and host community as entrepreneurs<br>6 new businesses operating under an umbrella organization<br>6 Photovoltaic Electrification Committees<br>1700 Solar Home Systems (SHS) | Solutions based on existing technologies and approaches (connection to grid)<br>Coordination efforts with other projects based on avoiding overlaps<br>Strong collaboration with the EEU (training, regularization of connections, project approval, economic contribution, etc.)<br>Focus on sustainability (great efforts placed on developing an appropriate methodology for trainings.<br>Innovative approach as refugees and host communities are targeted together and treated as equals<br>Solutions based on this approach have proven to be effective in different contexts<br>Sustainability built upon market-based approach (developed by thematic experts )<br>Coordination efforts to avoid negative impacts with other approaches (free delivery)<br>Strong sensitization component as this model is not common in field | Alianza Shire Phase II Action Document<br>Alianza Shire project description<br>MoU signed between the Ethiopian Electric Utility, The Agency for Refugees and Returnees Affairs, Stitching ZOA and AECID<br>Technical pre-design of Project<br>Alianza Shire General Brochure<br>Alianza Shire Technical Brochure |

**Table A2.** *Cont*

| Element | Description / Key Indicators | Additional Considerations | Supporting evidences |
|---|---|---|---|

| Steering Committee | The Steering Committee is made up of senior managers from each partnership member. The Steering Committee is responsible for guiding the strategic direction of the partnership and ensuring the necessary resources for the implementation of the Project. Strategic decisions affecting the Alianza Shire are made unanimously. | Periodic -bimonthly- face-to-face meetings Members are represented by high-level executives UNCHR Spain participates in every meeting (invited organization as per the Agreement) Decision making processes are mainly based on the resolution of critical aspects of project development although aspects such as innovation are also included Meetings are chaired and guided by itdUPM | Alianza Shire MoU (Pilot Project) Alianza Shire members Collaboration Agreement (Phase II Project) Project Management Plan Security Agreement Iberdrola CEO declarations OCHA director in UPM with SECIPIC |
| Management Committee | The Management Committee is responsible for deciding on the planning, design, implementation and monitoring and evaluation of the Phase II Project. It is composed of one or two individuals from the partners who are supported by groups of experts within their own organizations. Decisions on the Project are taken by consensus in the Management Committee. Each member shall consult beforehand with other members on any decision it takes concerning the Project that affects one or more members and/or the Project itself. | Periodic-monthly- face-to-face Members are represented by senior experts There is a strong monitoring component based on monthly reports. The Management Committee is the space in which the technical work of each organization is brought together, and shared operational and technical decisions taken. Meetings are chaired and guided by itdUPM | Alianza Shire members Collaboration Agreement (Phase II Project) Project Management Plan Security Agreement |
| Project Office (PO) | The PO is formed exclusively for the integration and coordination of necessary elements for the design, implementation, monitoring and evaluation of the Project. The Project Office is in charge of articulating these different work levels. It participates, through designated representatives and by invitation, in the different levels and decision-making bodies of the Alianza Shire with the objective of reporting on the status of the Project. It follows the guidelines established by the Management Committee. | Periodic -weekly - meetings Meetings are chaired and guided by itdUPM The PO is formed by four individuals and it is permanently present in the three locations of the project (Madrid, Addis Ababa and Shire) Although it is conceived as the Management Committee implementing branch, apart from the execution, some design and innovation aspects emerge from the PO. | Alianza Shire members Collaboration Agreement (Phase II Project) Project Management Plan |

**Table A2.** *Cont*

| Element | Description / Key Indicators | Additional Considerations | Supporting evidences |
| --- | --- | --- | --- |

| | | | |
|---|---|---|---|
| Communications Committee | It designs and executes the external communication and visibility strategy external to the partnership, devising communications protocols and briefs as well as organizing actions that promote visibility. Communications decisions prioritize all products generated by the Project and make visible the participation of all members of the partnership. | Periodic –tri-monthly- face-to-face meetings Members are represented by communication staff of the partners. The decision-making capacity of these representatives varies significantly between organizations Decisions are countersigned by the Steering Committee Special relevance of each member's visibility – including the presence of the corresponding logo. Meetings are chaired and guided by itdUPM UNCHR Spain participates in every meeting (invited organization as per the Agreement) | Alianza Shire members Collaboration Agreement (Phase II Project) Project Management Plan Communication Protocols |
| Organizations in the Field | The relationship with these organizations is theoretically based on the grants and MoUs signed with them. The participation and linkage of these organizations to the project differs substantially. Long-term relationships are fostered with the organizations that may be attached to the project once it has been finished (EEU and trained operators, for instance). Alianza Shire is permitted to operate in refugee camps and is widely recognized by authorities. Coordination with organizations that are not 'formally linked' to the project is boosted. One of these organizations, ZOA, participates in weekly coordination meetings and proactive participation and opinion sharing is sought | The main organizations in the field are the following: International NGOs (Implementing partners): ZOA, Norwegian Refugee Council, Don Bosco – Jugend Eine Welt Governmental bodies: Agency for Refugee and Returnees Affairs (ARRA), several ministries and regional bureaus. Local authorities: Woredas and Kebele administrations Refugee bodies: Refugee Central Committees, Women Associations, etc. Ethiopian Electric Utility: Regional, district and local levels UN organizations: United Nations High Commissioner for Refugees (UNHCR) | Alianza Shire Agreements with other partners |

**Appendix C**

Table A3. Statements related to SDGs and partnerships of the Alianza Shire members.

| Organization | Document | Statements related to SDGs and partnerships |
|---|---|---|
| AECID | Master Plan (2018–21) | "The Spanish International Cooperation will promote the construction and strengthening of partnerships with the different actors committed to achieving the SDGs […]. These will be promoted among the different actors of international cooperation, public, private and civil society, from Spain and our partner countries, to maximize synergies, complement resources, enrich learning and increase the development impact of interventions" and "the role of the private sector in our humanitarian action will be enhanced, where there is added value ". |
| Iberdrola | Mission statement (2016) | "Our mission is to create value in a sustainable way in the development of our activities for society, citizens, customers and shareholders, being the leading multinational group in the energy sector that provides quality service through the use of environmentally friendly energy sources […]" |
| Acciona | Mission statement (2016) | "Our mission is to be leaders in the creation, promotion and management of Infrastructure, Water, Services and Renewable Energy actively contributing to social welfare, sustainable development and the generation of value for our stakeholders" |
| Signify | Mission statement (2018) | Stresses the environmental dimension of their activities and the commitment of the organization to making "people's lives more comfortable and safe, more productive companies and more livable cities". |
| itdUPM | Statute (2016) | […] "promote the generation of awareness, knowledge and innovative solutions that contribute to the fulfillment of the Sustainable Development Goals and, thus, to human and sustainable development". |

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
