# Peer review of "Transformational Collaboration for the SDGs: The Alianza Shire’s Work to Provide Energy Access in Refugee Camps and Host Communities"

_sustainability, doi:10.3390/su12020539_

Round 1

Reviewer 1 Report

I have reviewed the manuscript entitled “Transformational collaboration for the SDGs: the Alianza Shire’s work to provide energy access in refugee camps and host communities” .  I have comments that I would like the authors to consider.

I feel that its relevance to this journal could be enhanced by some major changes of emphasis at strategic points.  

Line 91, 163, 165 “Sustainabiltiy” – Sustainability of what? Is it partnership sustainability? But more, on other (more substantive) aspects of sustainability would help (e.g. field of planning, development, environment). Also, please reference more articles from the journal Sustainability.  Since I’m not that sure whether the topic of this paper fits well with this journal, I think this is important for the author to define their research field in the aspect of sustainability and find relevant studies published in this journal. The introduction could be re-structured to highlight the contribution and the motivation of the paper. The authors should add more literature review on multi-stakeholder partnership and need to address what kind of methodology was used in the literature. Also, please address what is the gap in the literature found by the authors. The authors should discuss how the results can be interpreted in perspective of previous studies and of the working hypotheses. While there is some explanation about the background of the Alianza Shire but I believe more information is needed to both justify the case for study. Why this is a good case study for research and why the findings here may be transferable or generalizable to other cases? There are no sections “limitations of the study” and “future directions of the research”. These sections should be added too, given this is a research paper.

Author Response

Replies to Reviewers' comments

Dear reviewer,

Thank you for your thorough revision and useful suggestions. Please find below a brief explanation regarding the modifications made in this new version according to your comments.

Comment 1.1: “Line 91, 163, 165 “Sustainabiltiy” – Sustainability of what? Is it partnership sustainability? But more, on other (more substantive) aspects of sustainability would help (e.g. field of planning, development, environment). I think this is important for the author to define their research field in the aspect of sustainability and find relevant studies published in this journal.”

Applying to: all document/Introduction/The case of Alianza Shire

The vague use of term “Sustainability” has been substituted by more specific terms.

The introduction has been re-structured to better explain the research field in the aspect of sustainability that this manuscript addresses. A brief summary is provided below:

1.- Partnerships and transformation in UN Sustainability Agenda:

Transformations are urgently needed in a range of areas to achieve the SDGs.

Worrying lack of consensus in how to do this.

These transformations seek to exploit synergies between Goals to achieve multiple SDGs by organizing implementation around SDG interventions that generate significant co-benefits.

Collaboration can help to build more enduring governance structures.

Main added references: 1, 2, 3.

2.- Sustainability of refugee crisis response (mainly regarding energy access):

The way humanitarian agencies provide assistance is changing.

If market-based approaches and market systems are developed well, they can both foster more effective access to energy and provide a long-term self-sustaining environment that persists after aid is removed.

  1. Humanitarian agencies have held high hopes for contributions from the private sector.
  2. Inclusive planning and design approaches: co-develop sustainable interventions with specialist energy NGOs, development partners and the private sector. These collaborations are creating innovative energy delivery models that place the energy needs of refugees at the heart of interventions and deliver long-term sustainable energy services to affected populations.
  3. Policy making: Policies that are clear and coherent need to be put in place so that the private sector can be progressively engaged, and energy services can be scaled-up.

Main added references: 4, 5, 6, 7, 8, 9, 10, 11, 12

3.- Energy access project sustainability

The manuscript aims to present how an energy access project meets the main sustainability key success factors through analyzing the main Alianza Shire partnership CVC factors. Some sustainability key success factors are presented from the literature [13, 14], and can be categorized in three groups:

  1. a) User related factors: Technology and Quality, Payment Capacity and Management, Local Training and Awareness Campaigns.
  2. b) Energy service-related factors: Operation and Maintenance, Business Models, Stakeholders and Agreements.
  3. c) Policy and regulation related factors: Public Policies.

Main added references: 15, 16, 17, 13, 14.

Comment 1.2: “Please reference more articles from the journal Sustainability.”

Applying to: Introduction, Theoretical context of partnerships

The following articles from Sustainability have been referenced:

Horan, D. A New Approach to Partnerships for SDG Transformations. Sustainability 2019, 11 (18), 4947.

Schoon, M.; Cox, M. E. Collaboration, Adaptation, and Scaling: Perspectives on Environmental Governance for Sustainability. Sustainability 2018, 10 (3), 679.

Wigboldus, S.; Brouwers, J.; Snel, H. How a Strategic Scoping Canvas Can Facilitate Collaboration between Partners in Sustainability Transitions. Sustainability 2020, 12 (1), 168.

Prieto-Egido, I.; Simó-Reigadas, J.; Martínez-Fernández, A. Interdisciplinary Alliances to Deploy Telemedicine Services in Isolated Communities: The Napo Project Case. Sustainability 2018, 10 (7), 2288.

Eras-Almeida, A. A.; Fernández, M.; Eisman, J.; Martín, J. G.; Caamaño, E.; Egido-Aguilera, M. A. Lessons Learned from Rural Electrification Experiences with Third Generation Solar Home Systems in Latin America: Case Studies in Peru, Mexico, and Bolivia. Sustainability 2019, 11 (24), 7139.

Vallejo, B.; Oyelaran-Oyeyinka, B.; Ozord, N.; Bolo, M. Open Innovation and Innovation Intermediaries in Sub-Saharan Africa. Sustainability 2019, 11 (2), 392.

Warbroek, B.; Hoppe, T.; Coenen, F.; Bressers, H. The Role of Intermediaries in Supporting Local Low-Carbon Energy Initiatives. Sustainability 2018, 10 (7), 2450.

Comment 1.4: “The introduction could be re-structured to highlight the contribution and the motivation of the paper.”

Applying to: Introduction.

The introduction has been re-structured to highlight the motivation, research gap and contribution of the manuscript. A brief summary of the changes made is presented below:

Motivation: the need for transformative partnerships and the international consensus regarding the need for partnerships among different actors as a mean to innovate and provide an effective and durable response to refugee crisis have been added to the introduction and section on the theoretical context of partnerships.

Research gap: the “Collaborative Value Creation framework” (Austin and Seitanidi, 2012) is a promising insight on how to promote shared value in a multi-stakeholder partnership that takes into account two recognized approaches: transformational aspiration (1) and evolutionary perspective (2). According to Austin and Seitanidi (2012), there is a need to deepen our understanding of the factors that enable collaborative relationships to enter into a transformational stage. Given that these forms of partnership are both complex and poorly documented, they call for the development of in-depth case studies to explore their transformational potential further. Furthermore, the framework does not explicitly consider partnership facilitators, an emergent key role in multi-stakeholder practice, the potential and importance of which has been recognized by other related strands of research, such as Sustainability Transitions Theory.

Contribution: the study has both theoretical and practical implications. From a theoretical contributes to improving our understanding of the dynamics of transformational multi-stakeholder collaborations and clarifying the role of intermediaries by developing an illustrative in-depth case study of a multi-stakeholder partnership with a clear transformational aspiration, in a challenging context: access to energy in refugee camps.

From a practical point of view, the analysis of Alianza Shire through the CVC lens enables the extraction of valuable lessons and improvement opportunities for the initiative. Furthermore, our analysis shows how the CVC framework can be used as a tool for self-assessment within a continuous improvement cycle, a methodological exercise which has high potential for replicability in other multi-actor partnerships

Comment 1.5: “The authors should add more literature review on multi-stakeholder partnership and need to address what kind of methodology was used in the literature.”

Applying to: Theoretical context of partnerships, Research approach.

 A more thorough literature review on multi-stakeholder partnership has been added to the relevant sections. A brief summary is presented below:

  1. The CVC framework is based on the closely-related concept of creating shared value (CSV) which was developed by Porter & Kramer (2011) 18.
  2. Subsequent critiques of CSV 19
  3. Partnerships facilitators and SDG Agenda 1, 20, 21, 22, 23
  4. Other ‘value creation’ models or frameworks for the analysis: Collective-Conflictual Value Co-creation (24), Partnership Outcomes Assessing (25), Partnership Learning Loop (26), Assessing Partnership Performance (27), Strategic Scoping Canvas (3).
  5. Further CVC-related research: 28, 29, 30.

Comment 1.6: “Please address what is the gap in the literature found by the authors.”

Applying to: Introduction and Theoretical context of partnerships.

Additional comments on the gaps identified in the CVC framework have been added to the introduction and section on the theoretical context of partnerships. The authors note that the “Collaborative Value Creation framework” (Austin and Seitanidi, 2012) offers interesting insights on how to promote shared value in a multi-stakeholder partnership that takes into account two recognized approaches: transformational aspiration (1) and an evolutionary perspective (2). According to Austin and Seitanidi (2012), there is a need to deepen our understanding of the factors that enable collaborative relationships to enter into a transformational stage. Given that these forms of partnership are both complex and poorly documented, they call for the development of in-depth case studies to explore their transformational potential further. Furthermore, the framework does not explicitly consider partnership facilitators, an emergent key role in multi-stakeholder practice, the potential and importance of which has been recognized by other related strands of research, such as Sustainability Transitions Theory.

Comment 1.7: “The authors should discuss how the results can be interpreted in perspective of previous studies and of the working hypotheses.”

Applying to: Discussion.

Discussion has been re-structured to highlight the interpretation of results in perspective of the research objectives:

  1. Behavior of Alianza Shire through CVC analysis (evolutionary and transformational perspectives).
  2. Relevance of the facilitator in the partnership.

Comment 1.8: “I believe more information about Alianza Shire background is needed to both justify the case for study.”

Applying to: The case of the Alianza Shire , Results: CVC analysis of the Alianza Shire.

Further information has been added regarding the Alianza Shire. The focus of the Partnership is now mentioned more explicitly in the abstract and a brief account of the partnership has been included in the introduction.

Comment 1.9: “Why this is a good case study for research and why the findings here may be transferable or generalizable to other cases?”

Applying to: Introduction, Discussion, Research Approach

According to Austin and Seitanidi (2012), there is a need to deepen our understanding of the factors that enable collaborative relationships to enter into a transformational stage. Given that these forms of partnership are both complex and poorly documented, they call for the development of in-depth case studies to explore their transformational potential further.

The Alianza Shire is a unique case of a multi-actor partnership both in terms of the scope of the collaboration and the impact achieved since its creation in 2014. A wide variety of actors have been involved in designing and implementing innovative initiatives in order to provide energy access to refugees and host communities

Several paragraphs have been added to the Introduction, Research Approach and Discussion in order to highlight the relevance of case-study research in this context and the potential of the manuscript as a good case study for research.

Comment 1.10: “There are no sections “limitations of the study” and “future directions of the research”.

Applying to: Discussion.

The relevant sections have been added. A brief summary is presented below: have been added to the introduction and section on the theoretical context of partnerships.

  1. Limitations of the study:

Limitations to generalizability of results of a single-case study; a qualitative approach has been adopted and it would be desirable to enrich the appraisal in the future by developing a measure scale for the different components of the CVC framework; the study has  focused on two of the five components of CVC (spectrum and stages), not covering additional features such as the CV mindset, collaboration process and outcomes which were subsequently added to the model by Austin & Seitanidi.

  1. Future directions of the research: CVC applicability improvements. CVC + Sustainability Transitions. Using tools to align visions among all Alianza Shire stakeholders (as Strategic Scoping Canvas). Put into practice some Further CVC-related research, as connecting “end costumers” to create value with them (companies’ clients, university students, refugees). Explore the links between CVC and Sustainability Transitions Theory.

Additional references

(1)           Horan, D. A New Approach to Partnerships for SDG Transformations. Sustainability 2019, 11 (18), 4947. https://doi.org/10.3390/su11184947.

(2)           Schoon, M.; Cox, M. E. Collaboration, Adaptation, and Scaling: Perspectives on Environmental Governance for Sustainability. Sustainability 2018, 10 (3), 679. https://doi.org/10.3390/su10030679.

(3)           Wigboldus, S.; Brouwers, J.; Snel, H. How a Strategic Scoping Canvas Can Facilitate Collaboration between Partners in Sustainability Transitions. Sustainability 2020, 12 (1), 168. https://doi.org/10.3390/su12010168.

(4)           Refugees, U. N. H. C. for. Summary of the first Global Refugee Forum by the co-convenors https://www.unhcr.org/events/conferences/5dfa70e24/summary-first-global-refugee-forum-co-convenors.html (accessed Dec 29, 2019).

(5)           The Power to Respond. Nat Energy 2019, 4 (12), 989–989. https://doi.org/10.1038/s41560-019-0528-6.

(6)           Rosenberg-Jansen, S.; Tunge, T.; Kayumba, T. Inclusive Energy Solutions in Refugee Camps. Nat Energy 2019, 4 (12), 990–992. https://doi.org/10.1038/s41560-019-0516-x.

(7)           Boodhna, A.; Sissons, C.; Fullwood-Thomas, J. A Systems Thinking Approach for Energy Markets in Fragile Places. Nat Energy 2019, 4 (12), 997–999. https://doi.org/10.1038/s41560-019-0519-7.

(8)           Huber, S.; Mach, E. Policies for Increased Sustainable Energy Access in Displacement Settings. Nat Energy 2019, 4 (12), 1000–1002. https://doi.org/10.1038/s41560-019-0520-1.

(9)           Private Sector & Refugees—Pathways to Scale https://www.ifc.org/wps/wcm/connect/REGION__EXT_Content/IFC_External_Corporate_Site/Sub-Saharan+Africa/Resources/PSR-Pathways-to-Scale (accessed Dec 29, 2019).

(10)        Humanitarian Innovation: The State of the Art https://gsdrc.org/document-library/humanitarian-innovation-the-state-of-the-art/ (accessed Dec 29, 2019).

(11)        Off-grid Solar PV Power for Humanitarian Action: From Emergency Communications to Refugee Camp Micro-grids | Elsevier Enhanced Reader https://reader.elsevier.com/reader/sd/pii/S1877705814010480?token=0CB26865F14D18415D270FB4D88FEB76CFE933FB29113087DA58080296D004553C07398AA3BC95B71E29F178F2089471 (accessed Dec 29, 2019). https://doi.org/10.1016/j.proeng.2014.07.061.

(12)        Reports https://mei.chathamhouse.org/resources/reports (accessed Dec 29, 2019).

(13)        Prieto-Egido, I.; Simó-Reigadas, J.; Martínez-Fernández, A. Interdisciplinary Alliances to Deploy Telemedicine Services in Isolated Communities: The Napo Project Case. Sustainability 2018, 10 (7), 2288. https://doi.org/10.3390/su10072288.

(14)        Eras-Almeida, A. A.; Fernández, M.; Eisman, J.; Martín, J. G.; Caamaño, E.; Egido-Aguilera, M. A. Lessons Learned from Rural Electrification Experiences with Third Generation Solar Home Systems in Latin America: Case Studies in Peru, Mexico, and Bolivia. Sustainability 2019, 11 (24), 7139. https://doi.org/10.3390/su11247139.

(15)        Pade, C.; Mallinson, B.; Sewry, D. An Elaboration of Critical Success Factors for Rural ICT Project Sustainability in Developing Countries: Exploring the Dwesa Case. Journal of Information Technology Case and Application Research 2008, 10 (4), 32–55.

(16)        Martens, M. L.; Carvalho, M. M. Key Factors of Sustainability in Project Management Context: A Survey Exploring the Project Managers’ Perspective. International Journal of Project Management 2017, 35 (6), 1084–1102. https://doi.org/10.1016/j.ijproman.2016.04.004.

(17)        Økland, A. Gap Analysis for Incorporating Sustainability in Project Management. Procedia Computer Science 2015, 64, 103–109. https://doi.org/10.1016/j.procs.2015.08.469.

(18)        Porter, M. E. „& Kramer, MR (2011). Creating Shared Value. Harvard Business Review 2011, 89 (1/2), 62–77.

(19)        Crane, A.; Palazzo, G.; Spence, L. J.; Matten, D. Contesting the Value of “Creating Shared Value.” California management review 2014, 56 (2), 130–153.

(20)        Abbott, K. W.; Genschel, P.; Snidal, D.; Zangl, B. International Organizations as Orchestrators; Cambridge University Press, 2015.

(21)        Dodds, F. Multi-Stakeholder Partnerships: Making Them Work for the Post-2015 Development Agenda. Global Research Institute, available from:< www. un. org/en/ecosoc/newfunct/pdf15/2015partnerships_background_note. pdf 2015.

(22)        Bakhtiari, F. International Cooperative Initiatives and the United Nations Framework Convention on Climate Change. Climate policy 2018, 18 (5), 655–663.

(23)        Fowler, A.; Biekart, K. Multi‐stakeholder Initiatives for Sustainable Development Goals: The Importance of Interlocutors. Public Administration and Development 2017, 37 (2), 81–93.

(24)        Laamanen, M.; Skålén, P. Collective–Conflictual Value Co-Creation: A Strategic Action Field Approach. Marketing Theory 2015, 15 (3), 381–400.

(25)        Brinkerhoff, J. M. Assessing and Improving Partnership Relationships and Outcomes: A Proposed Framework. Evaluation and Program Planning 2002, 25 (3), 215–231. https://doi.org/10.1016/S0149-7189(02)00017-4.

(26)        Partnership Learning Loop http://www.learningloop.nl/ (accessed Dec 29, 2019).

(27)        Caplan, K.; Gomme, J.; Mugabi, J.; Stott, L. Assessing Partnership Performance : Understanding the Drivers for Success; Building Partnerships for Development (BPDWS): London, UK, 2007.

(28)        Austin, J. E.; Seitanidi, M. M. Creating Value in Nonprofit-Business Collaborations: New Thinking and Practice; John Wiley & Sons, 2014.

(29)        Mongelli, L.; Rullani, F. Creating Value in Nonprofit-Business Collaborations: New Thinking and Practice, by James E. Austin and M. May Seitanidi. San Francisco: John Wiley and Sons, 2014. 320 Pp. ISBN: 978-1118531136. Business Ethics Quarterly 2017, 27 (1), 151–154.

(30)        Moeller, S.; Ciuchita, R.; Mahr, D.; Odekerken-Schröder, G.; Fassnacht, M. Uncovering Collaborative Value Creation Patterns and Establishing Corresponding Customer Roles. Journal of service research 2013, 16 (4), 471–487.

Reviewer 2 Report

This is a strong case study article that makes very effective use of the Austin and Seitanidi (2012) Collaborative Value Creation (CVC) framework to explore the transformational dimensions of the Alianza Shire partnership. I have a few comments and suggestions below that I think would strengthen the article.

While the authors acknowledge both the usefulness and limitations of the CVC framework as a partnership analysis tool, its weaknesses are not explicitly stated in the abstract. This limitation of CVC should be noted upfront. There also does not appear to be any evidence of a wider, critical review of the literature on the closely-related concept of creating shared value (CSV) which was developed by Porter & Kramer (2011) and which Austin and Seitanidi (2012) acknowledge as being a significant building block for their CVC framework. Subsequent critiques of CSV by Crane et al (2014) among others would also make useful additions to the literature review for this article. Also to what extent have other authors reviewed and critiqued Austin and Seitanidi (2012)? What further CVC-related research and writing have these authors done subsequently?

In a related vein, I would like the authors to be more explicit in justifying their choice and use of the CVC framework in comparison with other possible options. Did they review and consider other ‘value creation’ models or frameworks for the analysis of the Alianza Shire partnership besides the integration and use of partnership brokering and facilitation concepts, frameworks and tools? For example, see Laamanem & Skålén (2015) who, although coming from a marketing theory standpoint, offer a very different collective-conflictual perspective on value co-creation that could potentially offer additional useful insights in analysing Alianza Shire and other multi-stakeholder collaborative arrangements. Laamanem & Skålén (2015) is one example of how the authors of the article might be able to acknowledge that there are other conceptual frameworks on value co-creation which offer alternative viewpoints and forms of analysis.

Additional Potential References

Crane, A., Palazzo, G., Spence, L. J., & Matten, D. (2014). Contesting the Value of “Creating Shared Value.” California Management Review, 56(2), 130–153. https://doi.org/10.1525/cmr.2014.56.2.130

Laamanen, M., & Skålén, P. (2015). Collective-Conflictual Value Co-creation: A strategic action field approach. Marketing Theory, 15(3), 381–400. https://doi.org/10.1177/1470593114564905

Porter, M. E., & Kramer, M. R. (2011). Shared Value: How to reinvent capitalism—and unleash a wave of innovation and growth. Harvard Business Review, 89(1/2), 62-77.

Author Response

Replies to Reviewers' comments

Dear reviewer,

Thank you for your thorough revision and useful suggestions. Please find below a brief explanation regarding the modifications made in this new version according to your comments. Please also note that the Introduction and Discussion have been re-structured to better address the comments of the other reviewer.

Comment 2.1: “CVC framework weaknesses are not explicitly stated in the abstract”.

Applying to: Abstract

The abstract has been modified to include a comment on this.

Comment 2.2: “Review of the literature on the closely-related concept of creating shared value (CSV)”.

Applying to: Theoretical context of partnerships

Information on the origins of the concept of shared value and some of the criticisms of this have been added to the introduction and Section 2 with further discussion of how this notion has been built upon by the CVC framework. Changes include:

  1. References to the work of Porter and Kramer (2011) and Shared Value with an explicit definition and information on its application: [for companies] policy and practices that enhance the competitiveness of a company while advancing the social conditions of the communities in which it operates. Identifying and connecting societal and economic progress. [for NGO and governments] Opportunity to be more effective thinking in value terms: considering benefits relative to costs rather than funds and effort expended.
  2. Mention is also made of criticisms of Shared Value such as the narrow private sector focus and tensions between social and economic goals and how the CVC framework has sought to address some of these concerns.

Comment 2.3: “Further CVC-related research”.

Applying to: Discussion

  1. Other parts of CVC: Spectrum and Stages considered, not covering CV mindset, collaboration process and outcomes.
  2. Connecting “end costumers” to create value with them: Uncovering Collaborative Value Creation Patterns and Establishing Corresponding Customer Roles.

Comment 2.4: “I would like the authors to be more explicit in justifying their choice and use of the CVC framework in comparison with other possible options. Did they review and consider other ‘value creation’ models or frameworks for the analysis of the Alianza Shire partnership”?

Applying to: Introduction, Research Approach

Other ‘value creation’ models or frameworks for the analysis: Collective-Conflictual Value Co-creation (24), Partnership Outcomes Assessing (25), Partnership Learning Loop (26), Assessing Partnership Performance (27), Strategic Scoping Canvas (3).

Choice of CVC framework: “Most partnership evaluations measure effectiveness by examining the short-term tangible outputs of joint activities. Such appraisals may bypass medium-term outcomes and long-term impact, and they do little to help us understand how partner relationships influence results”. CVC takes into account two relevant features transformational aspiration (1) and evolutionary perspective (2).

Additional references

(1)           Horan, D. A New Approach to Partnerships for SDG Transformations. Sustainability 2019, 11 (18), 4947. https://doi.org/10.3390/su11184947.

(2)           Schoon, M.; Cox, M. E. Collaboration, Adaptation, and Scaling: Perspectives on Environmental Governance for Sustainability. Sustainability 2018, 10 (3), 679. https://doi.org/10.3390/su10030679.

(3)           Wigboldus, S.; Brouwers, J.; Snel, H. How a Strategic Scoping Canvas Can Facilitate Collaboration between Partners in Sustainability Transitions. Sustainability 2020, 12 (1), 168. https://doi.org/10.3390/su12010168.

(4)           Refugees, U. N. H. C. for. Summary of the first Global Refugee Forum by the co-convenors https://www.unhcr.org/events/conferences/5dfa70e24/summary-first-global-refugee-forum-co-convenors.html (accessed Dec 29, 2019).

(5)           The Power to Respond. Nat Energy 2019, 4 (12), 989–989. https://doi.org/10.1038/s41560-019-0528-6.

(6)           Rosenberg-Jansen, S.; Tunge, T.; Kayumba, T. Inclusive Energy Solutions in Refugee Camps. Nat Energy 2019, 4 (12), 990–992. https://doi.org/10.1038/s41560-019-0516-x.

(7)           Boodhna, A.; Sissons, C.; Fullwood-Thomas, J. A Systems Thinking Approach for Energy Markets in Fragile Places. Nat Energy 2019, 4 (12), 997–999. https://doi.org/10.1038/s41560-019-0519-7.

(8)           Huber, S.; Mach, E. Policies for Increased Sustainable Energy Access in Displacement Settings. Nat Energy 2019, 4 (12), 1000–1002. https://doi.org/10.1038/s41560-019-0520-1.

(9)           Private Sector & Refugees—Pathways to Scale https://www.ifc.org/wps/wcm/connect/REGION__EXT_Content/IFC_External_Corporate_Site/Sub-Saharan+Africa/Resources/PSR-Pathways-to-Scale (accessed Dec 29, 2019).

(10)        Humanitarian Innovation: The State of the Art https://gsdrc.org/document-library/humanitarian-innovation-the-state-of-the-art/ (accessed Dec 29, 2019).

(11)        Off-grid Solar PV Power for Humanitarian Action: From Emergency Communications to Refugee Camp Micro-grids | Elsevier Enhanced Reader https://reader.elsevier.com/reader/sd/pii/S1877705814010480?token=0CB26865F14D18415D270FB4D88FEB76CFE933FB29113087DA58080296D004553C07398AA3BC95B71E29F178F2089471 (accessed Dec 29, 2019). https://doi.org/10.1016/j.proeng.2014.07.061.

(12)        Reports https://mei.chathamhouse.org/resources/reports (accessed Dec 29, 2019).

(13)        Prieto-Egido, I.; Simó-Reigadas, J.; Martínez-Fernández, A. Interdisciplinary Alliances to Deploy Telemedicine Services in Isolated Communities: The Napo Project Case. Sustainability 2018, 10 (7), 2288. https://doi.org/10.3390/su10072288.

(14)        Eras-Almeida, A. A.; Fernández, M.; Eisman, J.; Martín, J. G.; Caamaño, E.; Egido-Aguilera, M. A. Lessons Learned from Rural Electrification Experiences with Third Generation Solar Home Systems in Latin America: Case Studies in Peru, Mexico, and Bolivia. Sustainability 2019, 11 (24), 7139. https://doi.org/10.3390/su11247139.

(15)        Pade, C.; Mallinson, B.; Sewry, D. An Elaboration of Critical Success Factors for Rural ICT Project Sustainability in Developing Countries: Exploring the Dwesa Case. Journal of Information Technology Case and Application Research 2008, 10 (4), 32–55.

(16)        Martens, M. L.; Carvalho, M. M. Key Factors of Sustainability in Project Management Context: A Survey Exploring the Project Managers’ Perspective. International Journal of Project Management 2017, 35 (6), 1084–1102. https://doi.org/10.1016/j.ijproman.2016.04.004.

(17)        Økland, A. Gap Analysis for Incorporating Sustainability in Project Management. Procedia Computer Science 2015, 64, 103–109. https://doi.org/10.1016/j.procs.2015.08.469.

(18)        Porter, M. E. „& Kramer, MR (2011). Creating Shared Value. Harvard Business Review 2011, 89 (1/2), 62–77.

(19)        Crane, A.; Palazzo, G.; Spence, L. J.; Matten, D. Contesting the Value of “Creating Shared Value.” California management review 2014, 56 (2), 130–153.

(20)        Abbott, K. W.; Genschel, P.; Snidal, D.; Zangl, B. International Organizations as Orchestrators; Cambridge University Press, 2015.

(21)        Dodds, F. Multi-Stakeholder Partnerships: Making Them Work for the Post-2015 Development Agenda. Global Research Institute, available from:< www. un. org/en/ecosoc/newfunct/pdf15/2015partnerships_background_note. pdf 2015.

(22)        Bakhtiari, F. International Cooperative Initiatives and the United Nations Framework Convention on Climate Change. Climate policy 2018, 18 (5), 655–663.

(23)        Fowler, A.; Biekart, K. Multi‐stakeholder Initiatives for Sustainable Development Goals: The Importance of Interlocutors. Public Administration and Development 2017, 37 (2), 81–93.

(24)        Laamanen, M.; Skålén, P. Collective–Conflictual Value Co-Creation: A Strategic Action Field Approach. Marketing Theory 2015, 15 (3), 381–400.

(25)        Brinkerhoff, J. M. Assessing and Improving Partnership Relationships and Outcomes: A Proposed Framework. Evaluation and Program Planning 2002, 25 (3), 215–231. https://doi.org/10.1016/S0149-7189(02)00017-4.

(26)        Partnership Learning Loop http://www.learningloop.nl/ (accessed Dec 29, 2019).

(27)        Caplan, K.; Gomme, J.; Mugabi, J.; Stott, L. Assessing Partnership Performance : Understanding the Drivers for Success; Building Partnerships for Development (BPDWS): London, UK, 2007.

(28)        Austin, J. E.; Seitanidi, M. M. Creating Value in Nonprofit-Business Collaborations: New Thinking and Practice; John Wiley & Sons, 2014.

(29)        Mongelli, L.; Rullani, F. Creating Value in Nonprofit-Business Collaborations: New Thinking and Practice, by James E. Austin and M. May Seitanidi. San Francisco: John Wiley and Sons, 2014. 320 Pp. ISBN: 978-1118531136. Business Ethics Quarterly 2017, 27 (1), 151–154.

(30)        Moeller, S.; Ciuchita, R.; Mahr, D.; Odekerken-Schröder, G.; Fassnacht, M. Uncovering Collaborative Value Creation Patterns and Establishing Corresponding Customer Roles. Journal of service research 2013, 16 (4), 471–487.

Round 2

Reviewer 1 Report

You have adressed the issues raised to  some  extend. So I won't oppose the publication of this paper